# Learning Representations by Maximizing Mutual Information Across Views

**Philip Bachman**
Microsoft Research
phil.bachman@gmail.com

**R Devon Hjelm**
Microsoft Research, MILA
devon.hjelm@microsoft.com

**William Buchwalter**
Microsoft Research
wibuch@microsoft.com

## Abstract

We propose an approach to self-supervised representation learning based on maximizing mutual information between features extracted from multiple *views* of a shared *context*. For example, one could produce multiple views of a local spatio-temporal context by observing it from different locations (e.g., camera positions within a scene), and via different modalities (e.g., tactile, auditory, or visual). Or, an ImageNet image could provide a context from which one produces multiple views by repeatedly applying data augmentation. Maximizing mutual information between features extracted from these views requires capturing information about high-level factors whose influence spans multiple views – e.g., presence of certain objects or occurrence of certain events. Following our proposed approach, we develop a model which learns image representations that significantly outperform prior methods on the tasks we consider. Most notably, using self-supervised learning, our model learns representations which achieve 68.1% accuracy on ImageNet using standard linear evaluation. This beats prior results by over 12% and concurrent results by 7%. When we extend our model to use mixture-based representations, segmentation behaviour emerges as a natural side-effect. Our code is available online: https://github.com/Philip-Bachman/amdim-public.

## 1 Introduction

Learning useful representations from unlabeled data is a challenging problem and improvements over existing methods can have wide-reaching benefits. For example, consider the ubiquitous use of pre-trained model components, such as word vectors [Mikolov et al., 2013, Pennington et al., 2014] and context-sensitive encoders [Peters et al., 2018, Devlin et al., 2019], for achieving state-of-the-art results on hard NLP tasks. Similarly, large convolutional networks pre-trained on large supervised corpora have been widely used to improve performance across the spectrum of computer vision tasks [Donahue et al., 2014, Ren et al., 2015, He et al., 2017, Carreira and Zisserman, 2017]. Though, the necessity of pre-trained networks for many vision tasks has been convincingly questioned in recent work [He et al., 2018]. Nonetheless, the core motivations for unsupervised learning – namely minimizing dependence on potentially costly corpora of manually annotated data – remain strong.

We propose an approach to self-supervised representation learning based on maximizing mutual information between features extracted from multiple *views* of a shared *context*. This is analogous to a human learning to represent observations generated by a shared cause, e.g. the sights, scents, and sounds of baking, driven by a desire to predict other related observations, e.g. the taste of cookies. For a more concrete example, the shared context could be an image from the ImageNet training set,

and multiple views of the context could be produced by repeatedly applying data augmentation to the image. Alternatively, one could produce multiple views of an image by repeatedly partitioning its pixels into "past" and "future" sets, with the considered partitions corresponding to a fixed autoregressive ordering, as in Contrastive Predictive Coding [CPC, van den Oord et al., 2018]. The key idea is that maximizing mutual information between features extracted from multiple views of a shared context forces the features to capture information about higher-level factors (e.g., presence of certain objects or occurrence of certain events) that broadly affect the shared context.

We introduce a model for self-supervised representation learning based on local Deep InfoMax [DIM, Hjelm et al., 2019]. Local DIM maximizes mutual information between a *global* summary feature vector, which depends on the full input, and a collection of *local* feature vectors pulled from an intermediate layer in the encoder. Our model extends local DIM in three key ways: it predicts features across independently-augmented versions of each input, it predicts features simultaneously across multiple scales, and it uses a more powerful encoder. Each of these modifications provides improvements over local DIM. Predicting across independently-augmented copies of an input and predicting at multiple scales are two simple ways of producing multiple views of the context provided by a single image. We also extend our model to mixture-based representations, and find that segmentation-like behaviour emerges as a natural side-effect. Section 3 discusses the model and training objective in detail.

We evaluate our model using standard datasets: CIFAR10, CIFAR100, STL10 [Coates et al., 2011], ImageNet[1] [Russakovsky et al., 2015], and Places205 [Zhou et al., 2014]. We evaluate performance following the protocol described by Kolesnikov et al. [2019]. Our model outperforms prior work on these datasets. Our model significantly improves on existing results for STL10, reaching over 94% accuracy with linear evaluation and no encoder fine-tuning. On ImageNet, we reach over 68% accuracy for linear evaluation, which beats the best prior result by over 12% and the best concurrent result by 7%. We reach 55% accuracy on the Places205 task using representations learned with ImageNet data, which beats the best prior result by 7%. Section 4 discusses the experiments in detail.

## 2   Related Work

One characteristic which distinguishes self-supervised learning from classic unsupervised learning is its reliance on procedurally-generated supervised learning problems. When developing a self-supervised learning method, one seeks to design a problem generator such that models must capture useful information about the data in order to solve the generated problems. Problems are typically generated from prior knowledge about useful structure in the data, rather than from explicit labels.

Self-supervised learning is gaining popularity across the NLP, vision, and robotics communities – e.g., [Devlin et al., 2019, Logeswaran and Lee, 2018, Sermanet et al., 2017, Dwibedi et al., 2018]. Some seminal work on self-supervised learning for computer vision involves predicting spatial structure or color information that has been procedurally removed from the data. E.g., Doersch et al. [2015] and Noroozi and Favaro [2016] learn representations by learning to predict/reconstruct spatial structure. Zhang et al. [2016] introduce the task of predicting color information that has been removed by converting images to grayscale. Gidaris et al. [2018] propose learning representations by predicting the rotation of an image relative to a fixed reference frame, which works surprisingly well.

We approach self-supervised learning by maximizing mutual information between features extracted from multiples views of a shared context. For example, consider maximizing mutual information between features extracted from a video with most color information removed and features extracted from the original full-color video. Vondrick et al. [2018] showed that object tracking can emerge as a side-effect of optimizing this objective in the special case where the features extracted from the full-color video are simply the original video frames. Similarly, consider predicting how a scene would look when viewed from a particular location, given an encoding computed from several views of the scene from other locations. This task, explored by Eslami et al. [2018], requires maximizing mutual information between features from the multi-view encoder and the content of the held-out view. The general goal is to distill information from the available observations such that contextually-related observations can be identified among a set of plausible alternatives. Closely related work considers learning representations by predicting cross-modal correspondence [Arandjelović and Zisserman, 2017, 2018]. In concurrent work, Tian et al. [2019] propose applying global Deep InfoMax across

multiple image views constructed by splitting each source image into a pair of images comprising the *L* and *ab* channels of the LAB colorspace version of the source image. While the mutual information bounds in [Vondrick et al., 2018, Eslami et al., 2018] rely on explicit density estimation, our model uses the contrastive bound from CPC [van den Oord et al., 2018], which has been further analyzed by McAllester and Stratos [2018], and Poole et al. [2019].

Another line of prior work considers relating the information shared across multiple views of an input distribution. Earlier works in this direction, called *multiview learning*, were largely focused on methods based on Canonical Correlation Analysis, in settings where assumptions were imposed to permit a more rigorous analysis [Kakade and Foster, 2007, Sridharan and Kakade, 2008]. More recent work based on multiview CCA has extended these approaches beyond formally tractable settings, and is more akin to our own work [Wang et al., 2015]. Basing multiview learning on general mutual information, rather than correlation, seems to provide a broader foundation for future work.

Evaluating new self-supervised learning methods presents some challenges. E.g., performance gains may be largely due to improvements in model architectures and training practices, rather than advances in the self-supervised learning component. This point was addressed by Kolesnikov et al. [2019], who found massive gains in standard metrics when existing methods were reimplemented with up-to-date architectures and optimized to run at larger scales. When evaluating our model, we follow their protocols and compare against their optimized results for existing methods. Some potential shortcomings with standard evaluation protocols have been noted by Goyal et al. [2019].

# 3 Method Description

Our model, which we call Augmented Multiscale DIM (AMDIM), extends the local version of Deep InfoMax introduced by Hjelm et al. [2019] in several ways. First, we maximize mutual information between features extracted from independently-augmented copies of each image, rather than between features extracted from a single, unaugmented copy of each image.[2] Second, we maximize mutual information between multiple feature scales simultaneously, rather than between a single global and local scale. Third, we use a more powerful encoder architecture. Finally, we introduce mixture-based representations. We now describe local DIM and the components added by our new model.

## 3.1 Local DIM

Local DIM maximizes mutual information between global features $f_1(x)$, produced by a convolutional encoder $f$, and local features $\{f_7(x)_{ij} : \forall i, j\}$, produced by an intermediate layer in $f$. The subscript $d \in \{1, 7\}$ denotes features from the top-most encoder layer with spatial dimension $d \times d$, and the subscripts $i$ and $j$ index the two spatial dimensions of the array of activations in layer $d$.[3] Intuitively, this mutual information measures how much better we can guess the value of $f_7(x)_{ij}$ when we know the value of $f_1(x)$ than when we do not know the value of $f_1(x)$. Optimizing this *relative* ability to predict, rather than *absolute* ability to predict, helps avoid degenerate representations which map all observations to similar values. Such degenerate representations perform well in terms of absolute ability to predict, but poorly in terms of relative ability to predict.

For local DIM, the terms global and local uniquely define where features come from in the encoder and how they will be used. In AMDIM, this is no longer true. So, we will refer to the features that encode the data to condition on (global features) as *antecedent* features, and the features to be predicted (local features) as *consequent* features. We choose these terms based on their role in logic.

We can construct a distribution $p(f_1(x), f_7(x)_{ij})$ over (antecedent, consequent) feature pairs via ancestral sampling as follows: (i) sample an input $x \sim \mathcal{D}$, (ii) sample spatial indices $i \sim u(i)$ and $j \sim u(j)$, and (iii) compute features $f_1(x)$ and $f_7(x)_{ij}$. Here, $\mathcal{D}$ is the data distribution, and $u(i)/u(j)$ denote uniform distributions over the range of valid spatial indices into the relevant encoder layer. We denote the marginal distributions over per-layer features as $p(f_1(x))$ and $p(f_7(x)_{ij})$.

Given $p(f_1(x))$, $p(f_7(x)_{ij})$, and $p(f_1(x), f_7(x)_{ij})$, local DIM seeks an encoder $f$ that maximizes the mutual information $I(f_1(x); f_7(x)_{ij})$ in $p(f_1(x), f_7(x)_{ij})$.

## 3.2 Noise-Contrastive Estimation

The best results with local DIM were obtained using a mutual information bound based on Noise-Contrastive Estimation (NCE – [Gutmann and Hyvärinen, 2010]), as used in various NLP applications [Ma and Collins, 2018], and applied to infomax objectives by van den Oord et al. [2018]. This class of bounds has been studied in more detail by McAllester and Stratos [2018], and Poole et al. [2019].

We can maximize the NCE lower bound on $I(f_1(x); f_7(x)_{ij})$ by minimizing the following loss:

$$\mathbb{E}_{(f_1(x), f_7(x)_{ij})} \left[ \mathbb{E}_{N_7} \left[ \mathcal{L}_\Phi(f_1(x), f_7(x)_{ij}, N_7) \right] \right]. \tag{1}$$

The *positive sample* pair $(f_1(x), f_7(x)_{ij})$ is drawn from the joint distribution $p(f_1(x), f_7(x)_{ij})$. $N_7$ denotes a set of *negative samples*, comprising many "distractor" consequent features drawn independently from the marginal distribution $p(f_7(x)_{ij})$. Intuitively, the task of the antecedent feature is to pick its true consequent out of a large bag of distractors. The loss $\mathcal{L}_\Phi$ is a standard log-softmax, where the normalization is over a large set of *matching scores* $\Phi(f_1, f_7)$. Roughly speaking, $\Phi(f_1, f_7)$ maps (antecedent, consequent) feature pairs onto scalar-valued scores, where higher scores indicate higher likelihood of a positive sample pair. We can write $\mathcal{L}_\Phi$ as follows:

$$\mathcal{L}_\Phi(f_1, f_7, N_7) = -\log \frac{\exp(\Phi(f_1, f_7))}{\sum_{\tilde{f}_7 \in N_7 \cup \{f_7\}} \exp(\Phi(f_1, \tilde{f}_7))}, \tag{2}$$

where we omit spatial indices and dependence on $x$ for brevity. Training in local DIM corresponds to minimizing the loss in Eqn. 1 with respect to $f$ and $\Phi$, which we assume to be represented by parametric function approximators, e.g. deep neural networks.

## 3.3 Efficient NCE Computation

We can efficiently compute the bound in Eqn. 1 for many positive sample pairs, using large negative sample sets, e.g. $|N_7| \gg 10000$, by using a simple dot product for the matching score $\Phi$:

$$\Phi(f_1(x), f_7(x)_{ij}) \triangleq \phi_1(f_1(x))^\top \phi_7(f_7(x)_{ij}). \tag{3}$$

The functions $\phi_1/\phi_7$ non-linearly transform their inputs to some other vector space. Given a sufficiently high-dimensional vector space, in principle we should be able to approximate any (reasonable) class of functions we care about – which correspond to belief shifts like $\log \frac{p(f_7|f_1)}{p(f_7)}$ in our case – via linear evaluation. The power of linear evaluation in high-dimensional spaces can be understood by considering Reproducing Kernel Hilbert Spaces (RKHS). One weakness of this approach is that it limits the rank of the set of belief shifts our model can represent when the vector space is finite-dimensional, as was previously addressed in the context of language modeling by introducing mixtures [Yang et al., 2018]. We provide pseudo-code for the NCE bound in Figure 1.

When training with larger models on more challenging datasets, i.e. STL10 and ImageNet, we use some tricks to mitigate occasional instability in the NCE cost. The first trick is to add a weighted regularization term that penalizes the squared matching scores like: $\lambda(\phi_1(f_1(x))^\top \phi_7(f_7(x)_{ij}))^2$. We use NCE regularization weight $\lambda = 4\mathrm{e}^{-2}$ for all experiments. The second trick is to apply a soft clipping non-linearity to the scores after computing the regularization term and before computing the log-softmax in Eqn. 2. For clipping score $s$ to range $[-c, c]$, we applied the non-linearity $s' = c \tanh(\frac{s}{c})$, which is linear around 0 and saturates as one approaches $\pm c$. We use $c = 20$ for all experiments. We suspect there may be interesting formal and practical connections between regularization that restricts the variance/range/etc of scores that go into the NCE bound, and things like the KL/information cost in Variational Autoencoders [Kingma and Welling, 2013].

## 3.4 Data Augmentation

Our model extends local DIM by maximizing mutual information between features from augmented views of each input. We describe this with a few minor changes to our notation for local DIM. We construct the augmented feature distribution $p_\mathcal{A}(f_1(x^1), f_7(x^2)_{ij})$ as follows: (i) sample an input $x \sim \mathcal{D}$, (ii) sample augmented images $x^1 \sim \mathcal{A}(x)$ and $x^2 \sim \mathcal{A}(x)$, (iii) sample spatial indices $i \sim u(i)$ and $j \sim u(j)$, (iv) compute features $f_1(x^1)$ and $f_7(x^2)_{ij}$. We use $\mathcal{A}(x)$ to denote the

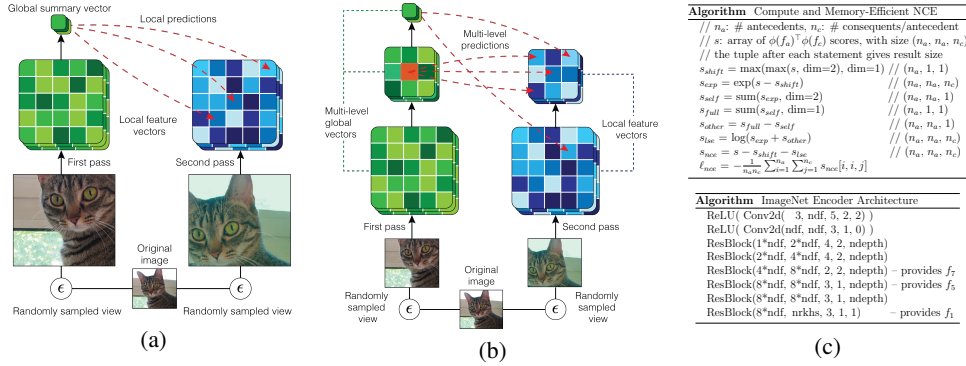

(a)             (b)             (c)

Figure 1: **(a)**: Local DIM with predictions across views generated by data augmentation. **(b)**: Augmented Multiscale DIM, with multiscale infomax across views generated by data augmentation. **(c)**-top: An algorithm for efficient NCE with minibatches of $n_a$ images, comprising one antecedent and $n_c$ consequents per image. For each true (antecedent, consequent) positive sample pair, we compute the NCE bound using all consequents associated with all other antecedents as negative samples. Our pseudo-code is roughly based on pytorch. We use dynamic programming in the log-softmax normalizations required by $\ell_{nce}$. **(c)**-bottom: Our ImageNet encoder architecture.

distribution of images generated by applying stochastic data augmentation to $x$. For this paper, we apply some standard data augmentations: random resized crop, random jitter in color space, and random conversion to grayscale. We apply a random horizontal flip to $x$ before computing $x^1$ and $x^2$.

Given $p_\mathcal{A}(f_1(x^1), f_7(x^2)_{ij})$, we define the marginals $p_\mathcal{A}(f_1(x^1))$ and $p_\mathcal{A}(f_7(x^2)_{ij})$. Using these, we rewrite the infomax objective in Eqn. 1 to include prediction across data augmentation:

$$\mathop{\mathbb{E}}_{(f_1(x^1), f_7(x^2)_{ij})} \left[ \mathop{\mathbb{E}}_{N_7} \left[ \mathcal{L}_\Phi(f_1(x^1), f_7(x^2)_{ij}, N_7) \right] \right], \tag{4}$$

where negative samples in $N_7$ are now sampled from the marginal $p_\mathcal{A}(f_7(x^2)_{ij})$, and $\mathcal{L}_\Phi$ is unchanged. Figure 1a illustrates local DIM with prediction across augmented views.

## 3.5 Multiscale Mutual Information

Our model further extends local DIM by maximizing mutual information across multiple feature scales. Consider features $f_5(x)_{ij}$ taken from position $(i, j)$ in the top-most layer of $f$ with spatial dimension $5 \times 5$. Using the procedure from the preceding subsection, we can construct joint distributions over pairs of features from any position in any layer like: $p_\mathcal{A}(f_5(x^1)_{ij}, f_7(x^2)_{kl})$, $p_\mathcal{A}(f_5(x^1)_{ij}, f_5(x^2)_{kl})$, or $p_\mathcal{A}(f_1(x^1), f_5(x^2)_{kl})$.

We can now define a family of $n$-to-$m$ infomax costs:

$$\mathop{\mathbb{E}}_{(f_n(x^1)_{ij}, f_m(x^2)_{kl})} \left[ \mathop{\mathbb{E}}_{N_m} \left[ \mathcal{L}_\Phi(f_n(x^1)_{ij}, f_m(x^2)_{kl}, N_m) \right] \right], \tag{5}$$

where $N_m$ denotes a set of independent samples from the marginal $p_\mathcal{A}(f_m(x^2)_{ij})$ over features from the top-most $m \times m$ layer in $f$. For the experiments in this paper we maximize mutual information from 1-to-5, 1-to-7, and 5-to-5. We uniformly sample locations for both features in each positive sample pair. These costs may look expensive to compute at scale, but it is actually straightforward to efficiently compute Monte Carlo approximations of the relevant expectations using many samples in a single pass through the encoder for each batch of $(x^1, x^2)$ pairs. Figure 1b illustrates our full model, which we call Augmented Multiscale Deep InfoMax (AMDIM).

## 3.6 Encoder

Our model uses an encoder based on the standard ResNet [He et al., 2016a,b], with changes to make it suitable for DIM. Our main concern is controlling receptive fields. When the receptive fields for features in a positive sample pair overlap too much, the task becomes too easy and the model performs worse. Another concern is keeping the feature distributions stationary by avoiding padding.

The encoder comprises a sequence of blocks, with each block comprising multiple residual layers. The first layer in each block applies mean pooling with kernel width $w$ and stride $s$ to compute a base output, and computes residuals to add to the base output using a convolution with kernel width $w$ and stride $s$, followed by a ReLU and then a $1 \times 1$ convolution, i.e. $w = s = 1$. Subsequent layers in the block are standard $1 \times 1$ residual layers. The mean pooling compensates for not using padding, and the $1 \times 1$ layers control receptive field growth. Exhaustive details can be found in our code: `https://github.com/Philip-Bachman/amdim-public`. We train our models using 4-8 standard Tesla V100 GPUs per model. Other recent, strong self-supervised models are non-reproducible on standard hardware.

We use the encoder architecture in Figure 1c when working with ImageNet and Places205. We use $128 \times 128$ input for these datasets due to resource constraints. The argument order for Conv2d is (input dim, output dim, kernel width, stride, padding). The argument order for ResBlock is the same as Conv2d, except the last argument (i.e. ndepth) gives block depth rather than padding. Parameters ndf and nrkhs determine encoder feature dimension and output dimension for the embedding functions $\phi_n(f_n)$. The embeddings $\phi_7(f_7)$ and $\phi_5(f_5)$ are computed by applying a small MLP via convolution. We use similar architectures for the other datasets, with minor changes to account for input sizes.

### 3.7 Mixture-Based Representations

We now extend our model to use mixture-based features. For each antecedent feature $f_1$, we compute a set of *mixture features* $\{f_1^1, ..., f_1^k\}$, where $k$ is the number of mixture components. We compute these features using a function $m_k$: $\{f_1^1, ..., f_1^k\} = m_k(f_1)$. We represent $m_k$ using a fully-connected network with a single ReLU hidden layer and a residual connection between $f_1$ and each mixture feature $f_1^i$. When using mixture features, we maximize the following objective:

$$\underset{f,q}{\text{maximize}} \ \underset{(x^1,x^2)}{\mathbb{E}} \left[ \frac{1}{n_c} \sum_{i=1}^{n_c} \sum_{j=1}^{k} \Big( q(f_1^j(x^1)|f_7^i(x^2)) \, s_{nce}(f_1^j(x^1), f_7^i(x^2)) + \alpha H(q) \Big) \right]. \quad (6)$$

For each augmented image pair $(x^1, x^2)$, we extract $k$ mixture features $\{f_1^1(x^1), ..., f_1^k(x^1)\}$ and $n_c$ consequent features $\{f_7^1(x^2), ..., f_7^{n_c}(x^2)\}$. $s_{nce}(f_1^j(x^1), f_7^i(x^2))$ denotes the NCE score between $f_1^j(x^1)$ and $f_7^i(x^2)$, computed as described in Figure 1c. This score gives the log-softmax term for the mutual information bound in Eqn. 2. We also add an entropy maximization term $\alpha H(q)$.

In practice, given the $k$ scores $\{s_{nce}(f_1^1, f_7^i), ..., s_{nce}(f_1^k, f_7^i)\}$ assigned to consequent feature $f_7^i$ by the $k$ mixture features $\{f_1^1, ..., f_1^k\}$, we can compute the optimal distribution $q$ as follows:

$$q(f_1^j|f_7^i) = \frac{\exp(\tau s_{nce}(f_1^j, f_7^i))}{\sum_{j'} \exp(\tau s_{nce}(f_1^{j'}, f_7^i))}, \quad (7)$$

where $\tau$ is a temperature parameter that controls the entropy of $q$. We motivate Eqn. 7 by analogy to Reinforcement Learning. Given the scores $s_{nce}(f_1^j, f_7^i)$, we could define $q$ using an indicator of the maximum score. But, when $q$ depends on the *stochastic* scores this choice will be overoptimistic in expectation, since it will be biased towards scores which are pushed up by the stochasticity (which comes from sampling negative samples). Rather than take a maximum, we encourage $q$ to be less greedy by adding the entropy maximization term $\alpha H(q)$. For any value of $\alpha$ in Eqn. 6, there exists a value of $\tau$ in Eqn. 7 such that computing $q$ using Eqn. 7 provides an optimal $q$ with respect to Eqn. 6. This directly relates to the formulation of optimal Boltzmann-type policies in the context of Soft Q Learning. See, e.g. Haarnoja et al. [2017]. In practice, we treat $\tau$ as a hyperparameter.

## 4 Experiments

We evaluate our model on standard benchmarks for self-supervised visual representation learning. We use CIFAR10, CIFAR100, STL10, ImageNet, and Places205. To measure performance, we first train an encoder using all examples from the training set (sans labels), and then train linear and MLP classifiers on top of the encoder features $f_1(x)$ (sans backprop into the encoder). The final performance metric is the accuracy of these classifiers. This follows the evaluation protocol described by Kolesnikov et al. [2019]. Our model outperforms prior work on these datasets.

| Method | ImageNet | Places205 |
|---|---|---|
| ResNet50v2 (sup) | 74.4 | 61.6 |
| AMDIM (sup) | 71.3 | 57.4 |
| Rotation | 55.4 | 48.0 |
| Exemplar | 46.0 | 42.7 |
| Patch Offset | 51.4 | 45.3 |
| Jigsaw | 44.6 | 42.2 |
| CPC - large | 48.7 | n/a |
| CPC - huge | 61.0 | n/a |
| CMC - large | 60.1 | n/a |
| AMDIM - small | 63.5 | n/a |
| AMDIM - large | **68.1** | 55.0 |

(a)

| | CIFAR10 (linear, MLP) | CIFAR100 (linear, MLP) |
|---|---|---|
| Highway Network | 92.28 | 67.61 |
| ResNet:101 | 93.58 | 74.84 |
| WideResNet:40-4 | 95.47 | 79.82 |
| AMDIM - small | 89.5, 91.4 | 68.1, 71.5 |
| AMDIM - large | 91.2, 93.1 | 70.2, 72.8 |

(b)

| | STL10 (linear, MLP) | ImageNet (linear, MLP) |
|---|---|---|
| AMDIM | 93.4, 93.8 | 61.7, 62.6 |
| +strong aug | 94.2, 94.5 | 62.7, 63.1 |
| −color jitter | 90.3, 90.6 | 57.7, 58.8 |
| −random gray | 88.3, 89.4 | 53.6, 54.9 |
| −random crop | 86.0, 87.1 | 53.2, 54.9 |
| −multiscale | 92.6, 93.0 | 59.9, 61.2 |
| −stabilize | 93.5, 93.8 | 57.2, 59.5 |

(c)

Table 1: **(a)**: Comparing AMDIM with prior results for the ImageNet and Imagenet→Places205 transfer tasks using linear evaluation. The competing methods are from: [Gidaris et al., 2018, Dosovitskiy et al., 2014, Doersch and Zisserman, 2017, Noroozi and Favaro, 2016, van den Oord et al., 2018, Hénaff et al., 2019]. The non-CPC results are from updated versions of the models by Kolesnikov et al. [2019]. The (sup) models were fully-supervised, with no self-supervised costs. The small and large AMDIM models had size parameters: (ndf=192, nrkhs=1536, ndepth=8) and (ndf=320, nrkhs=2560, ndepth=10). AMDIM outperforms prior and concurrent methods by a large margin. We trained AMDIM models for 150 epochs on 8 NVIDIA Tesla V100 GPUs. When we train the small model using a shorter 50 epoch schedule, it achieves 62.7% accuracy in 2 days on 4 GPUs.

Table 2: **(b)**: comparing AMDIM with fully-supervised models on CIFAR10 and CIFAR100, using linear and MLP evaluation. Supervised results are from [Srivastava et al., 2015, He et al., 2016a, Zagoruyko and Komodakis, 2016]. The small and large AMDIM models had size parameters: (ndf=128, nrkhs=1024, ndepth=10) and (ndf=256, nrkhs=2048, ndepth=10). AMDIM features performed on par with classic fully-supervised models.

Table 3: **(c)**: Results of single ablations on STL10 and ImageNet. The size parameters for all models on both datasets were: (ndf=192, nrkhs=1536, ndepth=8). We trained these models for 50 epochs, thus the ImageNet models were smaller and trained for one third as long as our best result (68.1%). We perform ablations against a baseline model that applies basic data augmentation which includes resized cropping, color jitter, and random conversion to grayscale. We ablate different aspects of the data augmentation as well as multiscale feature learning and NCE cost regularization. Our strongest results used the Fast AutoAugment augmentation policy from Lim et al. [2019], and we report the effects of switching from basic augmentation to stronger augmentation as "+strong aug". Data augmentation had the strongest effect by a large margin, followed by stability regularization and multiscale prediction.

On CIFAR10 and CIFAR100 we trained small models with size parameters: (ndf=128, nrkhs=1024, ndepth=10), and large models with size parameters: (ndf=320, nrkhs=1280, ndepth=10). On CIFAR10, the large model reaches 91.2% accuracy with linear evaluation and 93.1% accuracy with MLP evaluation. On CIFAR100, it reaches 70.2% and 72.8%. These are comparable with slightly older fully-supervised models, and well ahead of other work on self-supervised feature learning. See Table 2 for a comparison with standard fully-supervised models. On STL10, using size parameters: (ndf=192, nrkhs=1536, ndepth=8), our model significantly improves on prior self-supervised results. STL10 was originally intended to test semi-supervised learning methods, and comprises 10 classes with a total of 5000 labeled examples. Strong results have been achieved on STL10 via semi-supervised learning, which involves fine-tuning some of the encoder parameters using the available labeled data. Examples of such results include [Ji et al., 2019] and [Berthelot et al., 2019], which achieve 88.8% and 94.4% accuracy respectively. Our model reaches 94.2% accuracy on STL10 with linear evaluation, which compares favourably with semi-supervised results that fine-tune the encoder using the labeled data.

On ImageNet, using a model with size parameters: (ndf=320, nrkhs=2536, ndepth=10), and a batch size of 1008, we reach 68.1% accuracy for linear evaluation, beating the best prior result by over 12% and the best concurrent results by 7% [Kolesnikov et al., 2019, Hénaff et al., 2019, Tian et al., 2019]. Our model is significantly smaller than the models which produced those results and is reproducible on standard hardware. Using MLP evaluation, our model reaches 69.5% accuracy. Our linear and MLP evaluation results on ImageNet both surpass the original AlexNet trained end-to-end by a large margin. Table 3 provides results from single ablation tests on STL10 and ImageNet. We perform ablations on individual aspects of data augmentation and on the use of multiscale feature

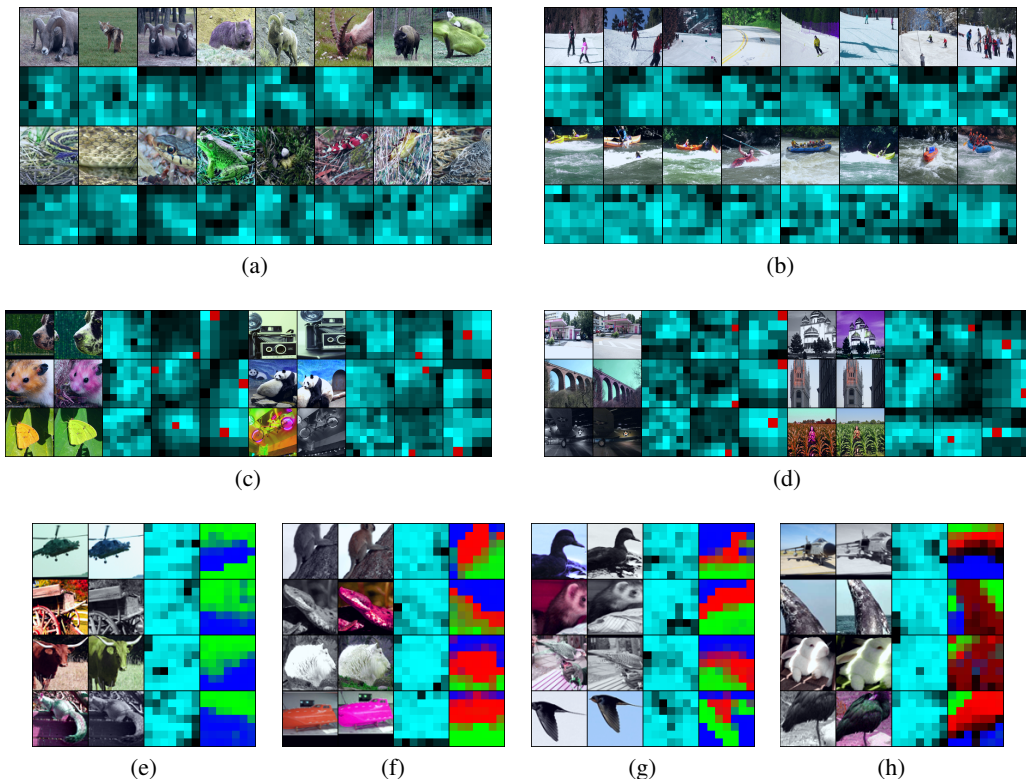

Figure 2: Visualizing behaviour of AMDIM. (a) and (b) combine two things – KNN retrieval based on cosine similarity between features $f_1$, and the matching scores (i.e., $\phi_1(f_1)^\top \phi_7(f_7)$) between features $f_1$ and $f_7$. (a) is from ImageNet and (b) is from Places205. Each left-most column shows a query image, whose $f_1$ was used to retrieve 7 most similar images. For each query, we visualize similarity between its $f_1$ and the $f_7$s from the retrieved images. On ImageNet, we see that good retrieval is often based on similarity focused on the main object, while poor retrieval depends more on background similarity. The pattern is more diffuse for Places205. (c) and (d) visualize the data augmentation that produces paired images $x^1$ and $x^2$, and three types of similarity: between $f_1(x^1)$ and $f_7(x^2)$, between $f_7(x^1)$ and $f_7(x^2)$, and between $f_5(x^1)$ and $f_5(x^2)$. (e, f, g, h): we visualize models trained on STL10 with 2, 3, 3, and 4 components in the top-level mixtures. For each $x^1$ (left) and $x^2$ (right), the mixture components were inferred from $x^1$ and we visualize the posteriors over those components for the $f_7$ features from $x^2$. We compute the posteriors as described in Section 3.7.

learning and NCE cost regularization. See Table 1 for a comparison with well-optimized results for prior and concurrent models. We also tested our model on an Imagenet→Places205 transfer task, which involves training the encoder on ImageNet and then training the evaluation classifiers on the Places205 data. Our model also beat prior results on that task. Performance with the transferred features is close to that of features learned on the Places205 data. See Table 1.

We include additional visualizations of model behaviour in Figure 2. See the figure caption for more information. Briefly, though our model generally performs well, it does exhibit some characteristic weaknesses that provide interesting subjects for future work. Intriguingly, when we incorporate mixture-based representations, segmentation behaviour emerges as a natural side-effect. The mixture-based model is more sensitive to hyperparameters, and we have not had time to tune it for ImageNet. However, the qualitative behaviour on STL10 is exciting and we observe roughly a 1% boost in performance with a simple bag-of-features approach for using the mixture features during evaluation.

## 5 Discussion

We introduced an approach to self-supervised learning based on maximizing mutual information between arbitrary features extracted from multiple views of a shared context. Following this approach,

we developed a model called Augmented Multiscale Deep InfoMax (AMDIM), which improves on prior results while remaining computationally practical. Our approach extends to a variety of domains, including video, audio, text, etc. E.g., we expect that capturing natural relations using multiple views of local spatio-temporal contexts in video could immediately improve our model.

Worthwhile subjects for future research include: modifying the AMDIM objective to work better with standard architectures, improving scalability and running on better infrastructure, further work on mixture-based representations, and examining (formally and empirically) the role of regularization in the NCE-based mutual information bound. We believe contrastive self-supervised learning has a lot to offer, and that AMDIM represents a particularly effective approach.

## Footnotes

[1]ILSVRC2012 version

[2]We focus on images in this paper, but the approach directly extends to, e.g.: audio, video, and text.

[3]$d$ refers to the layer's spatial dimension and should not be confused with its depth in the encoder.

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
