[Reviews · NeurIPS 2019]

Reviewer 1



The authors introduce AMDIM. Their aim is to maximize the mutual information between representations of different 'views' of the same `context'. They demonstrate the benefits of such a procedure on multiple tasks. The authors are inspired by Deep InfoMax (DIM). They claim to extend DIM by using independently augmented versions of each input and also by generating features across multiple scales. They also claim to use a more powerful encoder. The authors show an NCE based setup also demonstrate a mechanism to maximize multi scale mutual information. They end by showing mixture based representations and experiments related to the proposed techniques on a variety of data sets. Their main contribution seems to be the multiscale setup and the usage of data augmentation for generating multiple features of the same input. Thee mutual information between these features is then maximized. The f_1 features from the trained model are then used as inputs to linear and MLP classifiers. The extensions to DIM introduced by the authors are interesting and seem to be useful. They seem to be a logical step forward in the space of procedures that maximize mutual information for generating better data representations. There are, however, some questions that come to mind. -- It might be beneficial to add more details/explanations about the procedures being discussed in this work. For example, an explanation of the different steps in Fig 1c will make it easier to read. -- It might be beneficial to add a pictorial representation of the network architecture used to represent m_k -- Figure 3 might need a more detailed explanation. In say Fig 3a) what do the different rows represent. Furthermore, similarity between 2 entities of interest say A and B is often a single number. How is it being represented in Fig 3a) -- So often the multi-layer structure of a NN helps in capturing different levels of information at different levels. In this perspective, what does it exactly mean to have high mutual information between say f_5 and f_7 (or any such set of hidden layer features)? -- In principle DIM can be used with a similar encoder as AMDIM. It will be very interesting to see the performance of DIM in such a setup as a baseline result. This will help in gauging the importance/influence of the augmentation/multi-scale setup used in AMDIM -- In the context of views, what are the authors' thoughts on different views generated through say a different camera angle for images instead of processing done to produce augmented version of an image. Originality: The work is an extension of a known technique (DIM). It seems to be a logical step forward in the space of procedures that maximize mutual information for generating better data representations. Their main contributions seem to be the multiscale setup and the usage of data augmentation for generating multiple features of the same input. The mutual information between these features is then maximized. The f_1 features from the trained model are then used as inputs to linear and MLP classifiers. Quality: The authors do try to give the mathematical formulations behind their work and also provide empirical evaluations. Clarity : The manuscript seems to be written in a somewhat ok manner. There is, however room for improvement with certain places perhaps being a bit too concise. Significance: The work shows useful insights into the usefulness of modern Mutual Information maximization and can perhaps encourage more work in using such techniques for learning representations.

Reviewer 2



The authors present a self-supervised approach for visual representation learning. The approach is based on maximizing the mutual information between different "views" of the same data. In particular, the model is tasked with predicting features across augmented variations of each input, and across multiple scales. The training is based on a contrastive loss (InfoNCE) which itself a lower bound on the mutual information (if the positive and negative instances to contrast are sampled carefully). Using a large number of stabilisation and regularization tricks, the model can outperform existing algorithms on a variety of standard benchmarks. Most notably, it outperforms Alexnet trained end-to-end (under the linear evaluation protocol) and sets the new state-of-the-art on ImageNet. The paper is in general well written, but the clarity of exposition can be improved (details below). In contrast to CPC, the model can compute all necessary feature vectors in one forward pass which makes it more computationally attractive. On the other hand, the number of tricks required to make the training stable is stunning. Nevertheless, my score is based on the extremely strong empirical performance and view this work as an instance of "move fast and break things" and the necessary understanding of the success of such methods will follow at a later point. My score is not higher because this work doesn't even cite several highly relevant, information-theoretically backed frameworks for multi-view learning (e.g. [1, 2]). I have several questions and suggestions in the "improvements" section on the basis of which I will consider updating the score. [1] https://homes.cs.washington.edu/~sham/papers/ml/info_multi.pdf [2] https://homes.cs.washington.edu/~sham/papers/ml/regression_cca.pdf ======== Thanks a lot for the strong rebuttal. The new results are impressive and the added ablation tests quantify the importance of each method. I would urge the authors to establish a connection to [1,2], as it provides some theoretical understanding beyond the proposed multi-view approach.

Reviewer 3



In the local DIM method, the mutual information is maximized between a global summary feature vector, which depends on the full input, and a collection of local feature vectors pulled from an intermediate layer in the encoder. This paper extends this method in three ways: 1. It applies data augmentation to generate different views of an image. Then, it maximizes the mutual info between the two views, instead of a single view. 2. It proposes to maximize the mutual info between the features of any layers. 3. It uses a different more powerful encoder. Issues: - My main issue is the motivation behind each of the novelties of the paper. In subsections 3.4 and 3.5, the paper starts explaining the proposed approach without giving an explanation of why the proposed modifications to local DIM will be helpful. - Are these proposed modifications to local DIM really important? To get the answer, we need to look at the experimental results section. But then we found several issues in the experiments: 1) We do not see the results of local DIM (Hjelm et al. 2019) in the experiments. 2) There is no explanation about the results in table 1 in the text of the paper! This table and its results should be explained in detail such that we know which approach is better! 3) In the caption of table 1, it has been mentioned that "Data augmentation had the strongest effect by a large margin". What we see in table 1 is that multiscale has the largest effect, by a large margin. --------------- After reading the response of authors: The authors presented a new set of results that shows the method works well. But, my main issue is still clarity and the intuition behind each step of the proposed method. This issue has also been mentioned by Rev#2. It is also crucial to see the result of local DIM in all the tables (properly cited), not just in fig(b) of the response. For these reasons, I keep my score "below the threshold".

[Author Response · NeurIPS 2019]

| Method | ImageNet | Places205 |
|---|---|---|
| ResNet50v2 (sup) | 74.4 | 61.6 |
| AMDIM (sup) | 71.3 | 57.4 |
| Rotation | 55.4 | 48.0 |
| Exemplar | 46.0 | 42.7 |
| Patch Offset | 51.4 | 45.3 |
| Jigsaw | 44.6 | 42.2 |
| CPC - large | 48.7 | n/a |
| CPC - huge | 61.0 | n/a |
| CMC - large | 60.1 | n/a |
| AMDIM - small | 63.5 | n/a |
| AMDIM - large | **68.1** | 55.0 |

(a)

| | STL10 (linear, MLP) | ImageNet (linear, MLP) |
|---|---|---|
| AMDIM | 93.4, 93.8 | 61.7, 62.6 |
| +strong aug | 94.2, 94.5 | 62.7, 63.1 |
| −color jitter | 90.3, 90.6 | 57.7, 58.8 |
| −random gray | 88.3, 89.4 | 53.6, 54.9 |
| −random crop | 86.0, 87.1 | 53.2, 54.9 |
| −multiscale | 92.6, 93.0 | 59.9, 61.2 |
| −stabilize | 93.5, 93.8 | 57.2, 59.5 |
| −aug and multiscale | 74.2, 75.6 | 39.1, 41.3 |

(b)

| | STL10 (linear, MLP) | ImageNet (linear, MLP) |
|---|---|---|
| AMDIM | 93.6, 93.8 | 58.8, 60.9 |
| -color jitter | 83.7, 85.2 | 41.0, 44.0 |
| -resized crop | 88.4, 89.4 | 49.3, 52.6 |
| -multiscale | 91.6, 92.4 | 57.3, 60.0 |
| -stabilize | n/a, n/a | 57.0, 58.5 |
| -coordinates | 92.6, 93.3 | 58.8, 60.6 |

(c)

Figure 1: **(a)**: Updated main results. We made the model deeper and removed most batchnorm. These results are strong and reproducible. **(b)**: Updated ablation results. We split color-based augmentation into two parts: (i) color jitter and (ii) random grayscale. Models are the size of AMDIM-small from (a), but trained for fewer epochs due to resource constraints. **(c)**: Our original ablation results. Performance drops when we remove any of the components which AMDIM adds to DIM. When we remove data augmentation ("-color jitter" or "-resized crop") performance drops from 58.8% to 41.0% or 49.3%. When we remove multiscale prediction ("-multiscale") performance drops from 58.8% to 57.3%. Removing data augmentation causes a much larger performance drop than removing multiscale prediction. Note: "-color jitter" in (c) includes both types of color-based augmentation from (b).

**Response to Reviewers:**

We thank the reviewers for taking time to carefully review our paper and provide helpful feedback. We believe we can
address the reviewers' comments well, and will use them to improve the paper's clarity. We also have updated results
which strengthen the story and conclusions of our paper without requiring changes to the main technical content.

We made minor changes to our layer implementations and acquired access to infrastructure which allowed us to train
larger models in less time. This proved fruitful: using a larger encoder raised AMDIM's performance substantially
on ImageNet from 60.2% to 68.1%, and from 50.0% to 55.0% on the Places205 transfer task. See Fig. 1a and 1b
for more information. This outperforms prior results by 12% and concurrent results by 7%. We achieve these results
using a smaller encoder and over an order of magnitude less compute than the strongest concurrent results. AMDIM
now achieves over 62% on ImageNet after training for two days on four V100 GPUs, and over 68% after training for
seven days on eight V100 GPUs. The closest concurrent methods are trained on hundreds of TPUs and achieve slightly
over 61%. Training on 4-8 good GPUs is accessible to a wide range of researchers, and within the normal range for
competitive deep learning benchmarks. The code for reproducing our results is available online.

For a clearer comparison with the original version of DIM, we extend our ablation results to include simultaneous
ablation of data augmentation and multiscale prediction (see Fig. 1b). Removing both data augmentation and multiscale
prediction reverts AMDIM to the original DIM, but with our new encoder. Thus, these results compare AMDIM with
DIM while controlling for the encoder architecture. Adding data augmentation and multiscale prediction to DIM has
substantial benefits (+20% on ImageNet) and is necessary for achieving competitive results.

For R3: The claim that: "...multiscale has the largest effect, by a large margin." is incorrect, and could be due to
unclear notation in our original ablation results (see Fig. 1c). As described in the caption, removing either aspect of
data augmentation causes a larger performance drop than removing multiscale prediction. We will edit to clarify this.

For R1: Fig. 3a in the paper shows seven nearest images to a query image $x_q$ based on cosine similarity between $f_1$s, and
the similarities between $f_1(x_q)$ and each $f_7(x_r)$ from each retrieved image $x_r$. The similarities $\phi_1(f_1(x_q))^\top \phi_7(f_7(x_r))$
are visualized as a heatmap below each retrieved image $x_r$. The heatmaps match the spatial layout of the $7 \times 7$ grid of
$f_7$ features the encoder provides for each $x_r$. Intuitively, each heatmap shows which part of each $x_r$ AMDIM thinks
is most similar to $x_q$. We believe the natural transformations provided by multiple views of the same context from
different viewpoints will lead to improved features, and we're currently investigating this using video.

For R2: We use two tricks to stabilize training – i.e. regularizing the squared InfoNCE logits and soft clipping them
via tanh – which seems reasonable in the context of deep neural networks. AMDIM still works well without these
tricks, though removing them reduces performance (see Fig. 1b and 1c). Our updated model is simpler. It uses the
same regularization weight for all logits and does not use coordinate prediction, which we have removed from the paper.
Training stability resembles standard supervised learning, without the dramatic instability characteristic of GANs. We
use $f_1$, $f_5$, and $f_7$ because the other features available from our encoder increased compute cost without significantly
affecting performance. AMDIM performs well over a wide range of choices about encoder architecture, optimization
objective, and training hyperparams. We will add discussion of how CCA and multi-view learning relate to our work.
We share the same motivations as CCA-based multi-view learning, but we feel our formulation is more general and
better suited to use with large models and datasets. E.g., unlike [1, 2], we do not assume that each view contains
sufficient information for near-optimal prediction. E.g, we may maximize mutual info between patch-level features
which are individually weakly-predictive, but which contain complementary information about some shared cause.
Hand-wavily, correlations seem limiting compared to MI bounds which do not assume particular functional forms.

[Meta-Review · NeurIPS 2019]

The reviewers all like the paper and find the rebuttal convincing but there are sill some issue around the presentation and contents of the method. Two of three reviewers find this to be minor so all in all accept is recommended.